# Lipid Peroxidation in Subretinal Fluid: Some Light on the Prognosis Factors

**DOI:** 10.3390/biom11040514

**Published:** 2021-03-30

**Authors:** Francisco Bosch-Morell, Enrique García-Gen, Salvador Mérida, Mariola Penadés, Carmen Desco, Amparo Navea

**Affiliations:** 1Departamento Ciencias Biomédicas, Biomedical Research Institute, Universidad Cardenal Herrera-CEU, CEU Universities, Av. Seminario s/n, 46113 Valencia, Spain; fbosch@uchceu.es (F.B.-M.); enr.garcia.ce@ceindo.ceu.es (E.G.-G.); salvador.merida@uchceu.es (S.M.); mariola.penades@uchceu.es (M.P.); carmen.desco@uv.es (C.D.); 2Thematic Cooperative Health Network for Research in Ophthalmology (Oftared), Carlos III Health Institute, 28220 Madrid, Spain; 3FISABIO Oftalmología Médica, Retina Unit Pío Baroja 12, 46015 Valencia, Spain

**Keywords:** oxidative stress, lipid peroxidation, MDA, subretinal fluid, retinal detachment, myopia

## Abstract

The aim of this study was to identify a relation between the clinical characteristics and differences in lipid peroxidation in the subretinal fluid (SRF) of rhegmatogenous retinal detached patients by malondialdehyde (MDA) quantification. We collected 65 SRF samples from consecutive patients during scleral buckling surgery in rhegmatogenous retinal detachment (RRD) eyes. In addition to a complete ophthalmic evaluation, we studied the refractive status, evolution time, and the number of detached retinal quadrants to establish the extension of RRD. We studied the clinical aspects and oxidative stress and compared the characteristics among groups. We found that neither the evolution time of RRD nor the patients’ age correlated with the MDA concentration in the SRF. The MDA and the protein content of the SRF increased in the patients with high myopia and with more extended RRD. Our results suggest that oxidative imbalance was important in more extended retinal detachment (RD) and in myopic eyes and should be taken into account in the managing of these cases.

## 1. Introduction

Rhegmatogenous retinal detachment (RRD) is a severe and relatively frequent pathology. Its incidence is about one per 10,000 people yearly [1]; myopia increases the danger of detachment by 10-fold. One possible reason for this is that myopia leads to earlier vitreous liquefaction [2]. Despite good anatomic results after surgical treatment reaching 80–90%, visual recovery does not achieve the same rate: 69.6% with visual success in a review [3], with a median final visual acuity (VA) of 20/63, which is slightly further from a normal VA. Others have attributed this lack of total functional recovery after successfully retinal reattachment, to the programmed apoptosis of retinal cells [4,5].

Oxidative damage is a well-known mechanism of apoptosis induction. Both have been frequently founded in ocular diseases [6]. The vitreous humor of patients with retinal detachment has oxidative products in it [7,8], although not specifically with lipid peroxidation products (LPO), which have demonstrated highly toxic activity [9,10]. The nature of the subretinal fluid (SRF) present in RRD has not been thoroughly studied.

As SRF comes into direct contact with the cell membrane outer segments (OS) of the photoreceptors in the detached retina, lipid peroxidation research on this SRF can occupy a prominent place to study the pathogenesis of damage due to retinal detachment. Therefore, the relationship between lipid peroxidation and the parameters identified as key in this disease, such as age, refractive status, evolution time, and extension of RRD has not been studied in RRD [11].

Malondialdehyde (MDA) is one of the most well-known secondary products of lipid peroxidation (LPO), it has a half-life in physiological conditions that is superior to the extremely short half-lives of reactive oxygen species [12] and is a marker of peroxidative damage to cell membranes [13,14]. The aim of this study was to identify a relation between clinical characteristics and differences in LPO in the SRF of retinal detached patients through MDA quantification.

## 2. Materials and Methods

We studied a series of consecutive patients with rhegmatogenous retinal detachment (RRD) who underwent extraocular retinal detachment surgery in our clinic. We included phakic eyes, and pseudophakic eyes only if the refractive pre-cataract surgery status was known. The exclusion criteria were hyperopia over +0.5 diopters, proliferative vitreoretinopathy, complicated cataract surgery in the last 6 months, previous vitreoretinal surgery, diabetic retinopathy, or retinal vascular occlusion.

Patients underwent complete ophthalmological examination (best corrected visual acuity (BCVA), refractive status that permed us to separate in (E) emmetropes with a spherical equivalent between 0 and +0.5 diopters, low myopia (LM) between 0 and −5.95 D and high myopia (HM) above −6 D, anterior segment biomicroscopy, lens status determined, and indirect ophthalmoscope fundus exploration). The duration of RRD was considered based on the subjective symptoms referred to by the patients after asking them. All the patients signed a corresponding written informed consent. The Ethics Committee of the FISABIO-Oftalmología Hospital approved the study, which followed the Declaration of Helsinki.

During the retinal external detachment surgery, we obtained the SRF at the time of routine drainage. Once the scleral indentation was in place, but not tied, we made the scleral puncture at the point of the greatest projection of the detached retina. With a dry syringe, we slowly drew SRF from the perforation site without penetrating the subretinal space. The obtained samples (0.3–0.6 mL) were stored at −80 °C until used.

The MDA concentration in the SRF, a lipid peroxidation product, was measured using liquid chromatography (HPLC) and expressed as nanomols of MDA created per mg protein. Briefly, 0.1 mL of sample (or standard solutions prepared daily from 1,1,3,3-tetramethoxypropane) and 0.75 mL of working solution (0.37% thiobarbituric acid and 6.4% perchloric acid; 2/1, *v*/*v*) were mixed and heated to 95 °C for 1 h. After cooling (10 min in an ice water bath), the flocculent precipitate was removed by centrifugation at 3200× *g* for 10 min.

The supernatant was neutralized and filtered (0.22 µm) prior to injection in an ODS 5 µm column (250 × 4.6 mm). The mobile phase consisted in 50 mM phosphate buffer (pH 6.0) and methanol (58:42, *v*/*v*). Isocratic separation was performed at 1.0 mL/min flow (HPLC System, Waters, Milford, MA, USA) with detection at 532 nm (UV/VIS HPLC Detector 2475 Waters, Milford, MA, USA). The protein content was determined according to Lowry et al. [15] using bovine serum albumin as the standard.

Statistical analyses were performed with version 24.0 of the commercially available IBM SPSS software (IBM Corp. Released 2016. IBM SPSS Statistics for Windows, Version 24.0. IBM Corp., Armonk, NY, USA) and GraphPad Prism, version 7.04 for Windows (GraphPad Software, La Jolla, CA, USA). Values were expressed as the mean ± standard deviation (SD).

Comparisons between clinical characteristics were made by a one-way analysis of variance and false a discovery rate-adjusted *p*-value [FDR] of < 0.05. The ANOVA of the data found by the Brown—Forsythe test was carried out by taking either Tukey’s test as a post hoc test when the data indicated homogeneity in variances (*p* < 0.05) or a Dunnet T3 test when the variances differed. Statistical differences were set at *p* ≤ 0.01. Correlations were examined by a linear regression analysis and expressed as the Pearson’s correlation coefficient (*r*); *p* ≤ 0.01 was considered statistically significant.

## 3. Results

We studied 65 eyes from 65 patients (41 men and 24 women) aged 59.2 ± 11.1 years (within the 30–80 years range). Forty-three were emmetropic, 13 were low myopic, and 9 were high myopic eyes. Retinal detachment lasted for 2.0 ± 1.4 (0.3–6) weeks. Detached retina extension was 19 with one quadrant, 25 with two quadrants, 13 with three quadrants, and 8 eyes with four detached quadrants (Table 1).

The mean MDA concentration was 0.23 ± 0.10 µM (within the 0.06–0.47 µM range) and was 10.30 ± 5.18 (3.29–24.39) mg/mL for proteins (Table 2). There was a positive Pearson correlation (*r* = 0.629, *p* < 0.001) between MDA and the protein concentration for each patient (Figure 1A). The fact that no Pearson’s correlation was found among the MDA concentration in SRF, retinal detachment evolution time (Figure 1B, *r* = 0.032, *p* = 0.800), or patients’ age (Figure 1C, *r* = −0.064, *p* = 0.612) was noteworthy.

Interestingly, the MDA and protein concentration were significantly lower (Figure 2C,D, *p* < 0.01) when the retinal detachment (RD) extension was minor (only one affected quadrant) than for a broader RD (four affected quadrants). There was no connection between RD extension and degree of myopia (data not shown). Similarly, we also found the MDA and protein concentration to be significantly higher (both *p* < 0.01) in the high myopic group vs. the control group (Figure 2A,B). Thus, a positive correlation was determined between the degree of myopia and MDA (Figure 1D, *r* = −0.800, and *p* < 0.001).

## 4. Discussion

The downregulation of antioxidant defenses and an increase in free radical production cause the consequent oxidative damage of different cellular components, in particular lipids. LPO relates to several ocular pathologies [16]. Once lipid peroxides are unstable compounds, they tend to degrade rapidly in a variety of subproducts, which can accumulate in the SRF of those patients undergoing retinal detachment.

In this context, MDA is one of the most well-known secondary lipid peroxidation products [17], which is by far the most popular, has a longer half-life, and is a reliable indicator of oxidative damage to cells and tissues [18]. Once formed, MDA can be enzymatically metabolized (oxidation and decarboxylation to CO_2_ and H_2_O) or can react in vivo on cellular and tissular proteins and nucleic acids to form adducts. Far to neutralize its toxicity, these changes can induce either biomolecular damages or cell death [9,10], due to apoptosis, autophagy, and ferroptosis [19].

There are many toxic biological effects they can cause on proteins [20], such as pro-inflammatory alterations of components of the complement system factor H, the main regulator of the alternative pathway [21] and C3a, a proinflammatory complement component [22]; activation of a specific isoform of protein kinase C (PKC) [23]; MDA adducts with eElongation factor 2 (eEF2), which catalyzes the movement of the ribosome along the mRNA in protein synthesis, which could contribute to decline of protein synthesis [24]. MDA is an important contributor to DNA mutation and damage [25].

This study attempted to determine a relation between oxidative stress in patients with retinal detachment with different clinical findings. The retina is a target that is especially susceptible to lipid peroxidation due to its high metabolic activity, high oxygen pressure, and high Poly-Unsaturated Fatty Acid (PUFA) content in the membranes of the photoreceptors [26]. This suggests that the level of MDA found in SRF can be attributed mainly, although not solely, to the oxidative damage of the retina.

No statistical difference was observed for either the MDA or protein contents in relation to age or the retinal detachment evolution time in the study groups. This result indicates that neither of these parameters itself determines the degree of LPO subproducts present in SRF. This confirms our findings from a previous study conducted by our group using thiobarbituric acid reactive substances (TBAR), with a lower sensitivity and specificity [14,27].

The observed differences between the MDA concentrations in the high-myopic patients versus the other groups could relate to the accumulation of this biomarker in SRF as a product of LPO. Although our series is small and does not include a large number of myopic eyes, this coincides with the conclusions proposed by Romero et al. 1998 [14] and is consistent with the suggestion that oxidative damage plays a key role in myopia development, which significantly differs in patients with LM compared with those with HM [28]. A dysregulation in, or uncontrolled production of, oxidative products contributes to not only the initiation but also to the propagation of many pathological processes [29].

Regarding this matter, an association between increased oxidative stress in myopic eyes and early liquefaction of the vitreous body was previously described [30]. Despite the few high myopic patients in our study, which could have masked the results, they fell in line with those found by our group in previous works. Thus, this finding stresses the importance of oxidative stress on high myopia.

In the specific case of retinal detachment and myopia, the fact that the retina consumes the largest amount of oxygen in the body and is exposed to a continuous light stimulus may generate abundant free radicals [28,30,31], which may, in turn, alter lipids (among other substances) and aggravate pathological conditions. Kreissig et al. [32] found that the degree of myopia negatively influenced postoperative visual function after RRD surgery. Mohamed et al. [33] described a tendency toward re-detachment after surgery in myopic eyes. The accumulation of peroxided lipids in SRF might form part of it.

The other interesting clinical finding was the influence of the RRD size on the oxidative status of SRF but not on the evolution time. Presumably, the large amount of photoreceptor disc membranes exposed in the largest RRD would be the reason for this. As far as we know, this is the first time a positive relation has been found between oxidative stress markers and RRD size in SRF. An elevation in vitreous oxidative stress markers, associated with RRD extension, was previously described in two small series [5,8]. SRF and vitreous are not the same fluid.

SRF comes in direct contact with the detached external surface of the retina, where the membrane discs of photoreceptor cells are the main source of lipids susceptible to oxidative damage. As the diffusion of substances, including oxidative stress markers, occurs between the subretinal space and vitreous gel, the previously described findings in the vitreous correlate well with ours in SRF. Recently, high levels of MDA were also found in the tears of subjects with limited retinal detachment as in central serous, and related to the degree of disease activity, although this is a retinal detachment with a mechanism different from the rhegmatogenous and is typically much smaller [34].

Even so, our study was not designed to show the correlation between MDA labels and visual recovery after RRD; a specific study and a greater number of patients would be necessary to verify this hypothesis, which may be of interest in future research.

Despite the surgical techniques to treat RRD having improved and achieving high anatomical success rates, vision recovery remains a challenge. We know that retinal damage cannot be prevented, and our findings corroborate that other factors, such as the amount of retina involved, can influence the oxidative stress-induced damage.

The suggestion made by others [7] of using antioxidants to improve better visual recovery choices after RRD are of interest. Gao et al. [35] demonstrated that blocking oxidative stress in experimental retinal detachment improved the apoptosis of photoreceptors. This may be an interesting future avenue of research to attempt to prevent the apoptosis of photoreceptors and, therefore, to improve the vision of patients with retinal detachment.

Recently, different treatments have achieved neuroprotective effects associated with a decrease in oxidative stress in various experimental animal models of DR [36,37,38]. Along the same lines, the inhibition of certain mitochondrial pathways mediated by oxidative stress preserves photoreceptors after retinal detachment [39]. This would be particularly important in the case of highly myopic eyes due to the impaired oxidative stress situation in the eyes of these patients [28,40,41] where it has also been reported that there is a decrease in the antioxidant proteins in the vitreous humor of pathological myopia patients undergoing retinal surgical treatment (among others for rhegmatogenous retinal detachment) compared with in controls [42].

In light of these matters, our results for MDA and the degree of myopia (Figure 1D) and retinal detachment extension (Figure 2C) suggest that patients with these clinical conditions have a worse oxidative balance, which results in greater damage during and after detachment and could be responsible for a worse functional (visual) prognosis. The results of this study not only have the value of MDA as a biomarker but would also support new treatments with the aim of increasing the success of retinal cell survival during detachment.

## Figures and Tables

**Figure 1 biomolecules-11-00514-f001:**
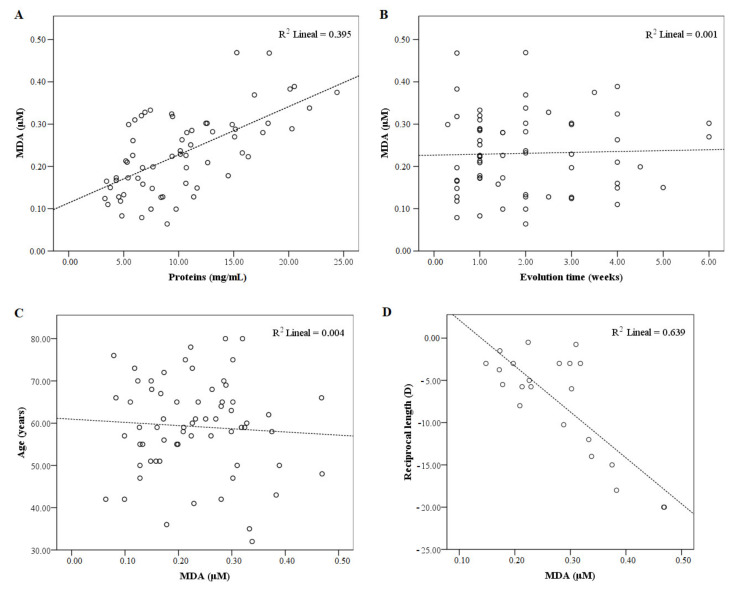
Correlations between the malondialdehyde (MDA) and the main studied parameters. The scatter plot showing Pearson’s correlation between: (**A**) MDA and protein concentration, *r* = 0.629, *p* < 0.001, and *n* = 65; (**B**) MDA and retinal detachment evolution time, *r* = 0.032, *p* = 0.800, and *n* = 65; (**C**) MDA and patients’ age, *r* = −0.064, *p* = 0.612, and *n* = 65; and (**D**) MDA and degree of myopia, *r* = −0.800, *p* < 0.001, and *n* = 22.

**Figure 2 biomolecules-11-00514-f002:**
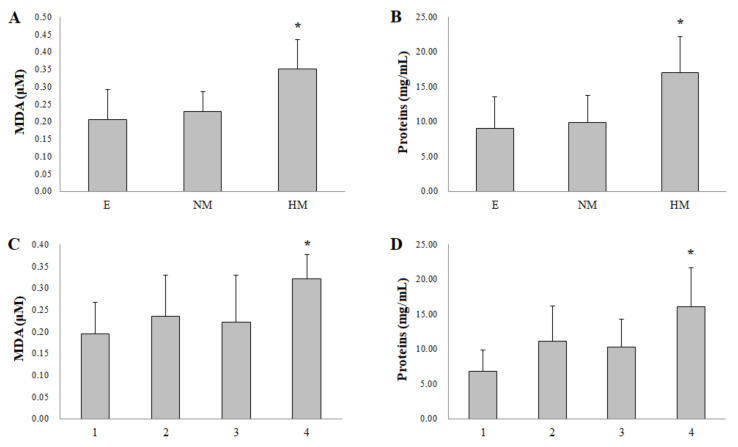
Oxidative stress parameter malondialdehyde (MDA) and protein concentration in the collected subretinal fluid. (**A**,**B**), comparison among the three refractive groups: E: emmetropics (*n* = 43); LM: Low myopic (*n* = 13); and HM: High myopic (*n* = 9). * *p* < 0.01 vs. the E group. (**C**,**D**), comparison by retinal detachment extension (in number of quadrants). * *p* < 0.01 vs. 1 quadrant.

**Table 1 biomolecules-11-00514-t001:** The main features of the 65 patients (41 men and 24 women) included in this study.

Age (Years)	59.2 ± 11.1 *	30–80 **		
Evolution time (weeks)	2.0 ± 1.4 *	0.3–6 **		
Extension retinal detachment (RD) (quadrants)	Q1 19	Q2 25	Q3 13	Q4 8
Refractive classification	E 43	LM 13	HM 9	

* Mean ± standard deviation (SD) ** range, Q: quadrants, E: emmetropic, LM: Low myopic, and HM: high myopic.

**Table 2 biomolecules-11-00514-t002:** Concentration in subretinal fluid.

	Mean Value	Range
Malondialdehyde (MDA) (µM)	0.23 ± 0.10	0.06 to 0.47
Protein (mg/mL)	10.30 ± 5.18	3.29 to 24.39

## Data Availability

Not applicable.

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
