# Peer review of "Lipid Peroxidation in Subretinal Fluid: Some Light on the Prognosis Factors"

_biomolecules, 2021, doi:10.3390/biom11040514_

Round 1

Reviewer 1 Report

In this article, the authors determined the malondialdehyde (MDA) levels in the subretinal fluid (SFR) specimens from rhegmatogenous retinal detachment (RRD) patients. The authors clarified that MDA levels in SFR correlated well with the degrees of myopia in patients and hypothesized that oxidation stress should play a crucial role in the RRD pathogenesis and evolution, and therefore that the sequelae by oxidation stress might be involved in the progression of medical conditions, likely myopia conditions. However, the authors should include additional experiments or comprehensive discussions as the following points for improving the article.

(1) The MDA level in the SRF can be use as biomarker of peroxidation damage to the cells, or retinal neurons. However, MDA is one of secondary products of lipid peroxidation (LPO). How could MDA levels in SRF reflect the progression of retinal damage? Is there any discrepancy between MDA levels and the medical conditions?  MDA levels in SRF do not directly reflect oxidation levels in retina, do they?

(2) The authors should discuss on how MDA behaves in SRF such as the half life of MDA in SRF, the stability of MDA in SRF and the toxicity of MDA. The reader could easily figure out the progression of RRD pathogenesis and how oxidation stress might be relevant to myopia conditions. The readers would like to know the detail of the behaviors of MDA in SRF, because the indication by the authors is so promising and the therapeutic approach to relief oxidation stress in retina can pave the avenue to enable future design and development of novel therapeutic candidates.

(3) The key figure in the article seems to be Figure 1. Does Fig.1D include all data? Fig.1D illustrates a good correlation between MDA levels and the progression of RRD.

However, the data number seems to be extremely different from other panels (A), (B) and (C). Why do the data number alter among panels? The data should be classified in detail, according to patient conditions.  

The reviewer makes sure that these major comments might bring a great improvement in the quality of the article.

Author Response

  • The MDA level in the SRF can be use as biomarker of peroxidation damage to the cells, or retinal neurons. However, MDA is one of secondary products of lipid peroxidation (LPO). How could MDA levels in SRF reflect the progression of retinal damage? Is there any discrepancy between MDA levels and the medical conditions?, MDA levels in SRF do not directly reflect oxidation levels in retina, do they?

There are a long and increasing evidence in the use of several markers of oxidative stress as an adequate clinical marquer (Frijhoff J, 2015). With no doubt, some of the most valued in ocular diseases are lipid peroxidation products such as MDA (Njie-Mbye YF, 2013). MDA has been proposed as biomarker in glaucoma (Benoist d'Azy C, 2016; Li S, 2020), age-related macular degeneration (Pinna A, 2020), diabetic retinopathy (Pan HZ, 2008), cataracts (AteĹź NA, 2004; Chitra PS, 2020). (We have not added this paragraph to the text in order not to increase the total length and quotes)

Frijhoff J, Winyard PG, Zarkovic N, Davies SS, Stocker R, Cheng D, Knight AR, Taylor EL, Oettrich J, Ruskovska T, Gasparovic AC, Cuadrado A, Weber D, Poulsen HE, Grune T, Schmidt HH, Ghezzi P. Clinical Relevance of Biomarkers of Oxidative Stress. Antioxid Redox Signal. 2015 Nov 10;23(14):1144-70. doi: 10.1089/ars.2015.6317. Epub 2015 Oct 26. PMID: 26415143.

Njie-Mbye YF, Kulkarni-Chitnis M, Opere CA, Barrett A, Ohia SE. Lipid peroxidation: pathophysiological and pharmacological implications in the eye. Front Physiol. 2013 Dec 16;4:366. doi: 10.3389/fphys.2013.00366. PMID: 24379787

Benoist d'Azy C, Pereira B, Chiambaretta F, Dutheil F. Oxidative and Anti-Oxidative Stress Markers in Chronic Glaucoma: A Systematic Review and Meta-Analysis. PLoS One. 2016 Dec 1;11(12):e0166915. doi:10.1371/journal.pone.0166915. PMID: 27907028.

Li S, Shao M, Li Y, Li X, Wan Y, Sun X, Cao W. Relationship between Oxidative Stress Biomarkers and Visual Field Progression in Patients with Primary Angle Closure Glaucoma. Oxid Med Cell Longev. 2020 Aug 5;2020:2701539. doi:10.1155/2020/2701539. PMID: 32831992

Pinna A, Boscia F, Paliogiannis P, Carru C, Zinellu A. MALONDIALDEHYDE LEVELS

IN PATIENTS WITH AGE-RELATED MACULAR DEGENERATION: A Systematic Review and Meta-analysis. Retina. 2020 Feb;40(2):195-203. doi: 10.1097/IAE.0000000000002574.

PMID: 31972788.

Pan HZ, Zhang H, Chang D, Li H, Sui H. The change of oxidative stress products in diabetes mellitus and diabetic retinopathy. Br J Ophthalmol. 2008 Apr;92(4):548-51. doi: 10.1136/bjo.2007.130542. PMID: 18369071.

AteĹź NA, Yildirim O, Tamer L, Unlü A, Ercan B, MuĹźlu N, Kanik A, Hatungil R, Atik U. Plasma catalase activity and malondialdehyde level in patients with cataract. Eye (Lond). 2004 Aug;18(8):785-8. doi: 10.1038/sj.eye.6700718. PMID: 15295623.

Chitra PS, Chaki D, Boiroju NK, Mokalla TR, Gadde AK, Agraharam SG, Reddy GB. Status of oxidative stress markers, advanced glycation index, and polyol pathway in age-related cataract subjects with and without diabetes. Exp Eye Res. 2020 Nov;200:108230. doi: 10.1016/j.exer.2020.108230. Epub 2020 Sep 12. PMID: 32931824.

We have modified the document to express that:

Discussion Pag 6 line 2 “The retina is a target especially susceptible to lipid peroxidation because of its high metabolic activity, high oxygen pressure and high PUFAs (Poly-Unsaturated Fatty Acids) content in the membranes of the photoreceptors [26], this suggests that the level of MDA found in SRF can be attributed mainly, although not only, to the oxidative damage of the retina”.

Discussion pag 7 line 23  Our results for MDA and degree of myopia (Fig. 1 D) and retinal detachment extension (Fig. 2C) suggest that patients with these clinical conditions have a worse oxidative balance, that mean greater damage during and after detachment and could be responsible for a worse functional (visual) prognosis. The results of this study not only have the value of MDA as a biomarker but would also support new treatments with the aim of increasing the success of retinal cell survival during detachment.

Pag 7, line 3our study has not been designed to show that, a specific study and a greater number of patients would be necessary to verify this hypothesis, that seems to be an interesting line of work.”

Discussion pag 7 lin 15 , “Recently, different treatments have achieved neuroprotective effects associated with a decrease in oxidative stress in various experimental animal models of DR [36,37,38]. Along the same lines, the inhibition of certain mitochondrial pathways mediated by oxidative stress preserves photoreceptors after retinal detachment [39]. This would be especially important in the case of high myopic eyes due to the impaired oxidative stress situation in the eyes of these patients [28,40,41], where it has also been reported that there is a decrease in antioxidant proteins in vitreous humor of pathological myopia patients undergoing retinal surgical treatment (among others for rhegmatogenous retinal detachment) than in controls [42]”.

(2) The authors should discuss on how MDA behaves in SRF such as the half-life of MDA in SRF, the stability of MDA in SRF and the toxicity of MDA. The reader could easily figure out the progression of RRD pathogenesis and how oxidation stress might be relevant to myopia conditions. The readers would like to know the detail of the behaviors of MDA in SRF, because the indication by the authors is so promising and the therapeutic approach to relief oxidation stress in retina can pave the avenue to enable future design and development of novel therapeutic candidates

Introduction .pag 2, line 6it has a half-life in physiological conditions very superiors to the extremely short half-lives of reactive oxygen species [12] and is a marker of peroxidative damage to cell membranes [13,14].

Discussion pag 5, line 7   “it has a longer half-life and is a reliable indicator of oxidative damage to cells and tissues [183]. Once formed, MDA can be enzymatically metabolized (oxidation and decarboxylation to CO2 and H2O) or can react in vivo on cellular and tissular proteins and nucleic acids to form adducts. Far to neutralize its toxicity, these changes can induce either biomolecular damages or cell death [9,10], because apoptosis, autophagy and ferroptosis [19]. There are many toxic biological effects that they can cause on proteins (GÄ™gotek A, 2019) such as pro-inflammatory alterations of components of the complement system  factor H, main regulator of the alternative pathway (Weismann D, 2011) and C3a, proinflammatory complement component (Veneskoski M, 2011); activation of a specific isoform of protein kinase C (PKC) (Kharbanda KK, 2002); MDA adducts with eElongation factor 2 (eEF2), which catalyzes the movement of the ribosome along the mRNA in protein synthesis could contribute to decline of protein synthesis (Argüelles S, 2013);… Furthermore, MDA is an important contributor to DNA mutation and damage (Gentile F, 2017).”

Discussion .pag 7 line 15: “Recently, different treatments have achieved neuroprotective effects associated with a decrease in oxidative stress in various experimental animal models of DR [36,37,38]. Along the same lines, the inhibition of certain mitochondrial pathways mediated by oxidative stress preserves photoreceptors after retinal detachment [39]. This would be especially important in the case of high myopic eyes due to the impaired oxidative stress situation in the eyes of these patients [28,40,41], where it has also been reported that there is a decrease in antioxidant proteins in vitreous humor of pathological myopia patients undergoing retinal surgical treatment (among others for rhegmatogenous retinal detachment) than in controls [42]”.

(3) The key figure in the article seems to be Figure 1. Does Fig.1D include all data? Fig.1D illustrates a good correlation between MDA levels and the progression of RRD. However, the data number seems to be extremely different from other panels (A), (B) and (C). Why do the data number alter among panels? The data should be classified in detail, according to patient conditions.  

Figure 1D exclusively includes myopic patients (n = 22; NM and HM groups). The intention of the authors is to determine if there was a relationship between MDA and degree of myopia, therefore emmetropes patients (group E) have been excluded. It has been added both in the text and in the figure caption. We apologize as it was not properly described.

Pag 4 foot of figure 1 n=22 only myopic eyes (HM and LM).

Pag 4 line 7 when excluding emmetropes

Also we have sent the manuscript to the English service

The reviewer makes sure that these major comments might bring a great improvement in the quality of the article.

We agree, and we thank the reviewer for his valuable help, 

Reviewer 2 Report

Description of the results is not clear. 65 eyes were studied, 43 served as control, thus it means that majority of studied eyes were control eyes. How control group was provided? Was scleral buckling surgery performed in normal eyes with retinal detachment? It should be clearly stated that control eyes were emmetropic eyes with retinal detachment. There were only 9 high myopic eyes and 13 low myopic eyes. These numbers are low. How did you assess the number of subjects in each group?  In the abstract conclusion is not supported by the results. It is not clearly described what groupd of patients are examined.

It is later written: “Detached retina extension was 19, 25, 13 91 and 8 eyes for one, two, three and four detached quadrants, respectively”. This sentence is not easy to be understood.

Statistical differences were set at p ≤ 0.01 why not 0.05?

Abbreviations should be described when describing figures (for example MDA in figure 1 description).

In the introduction the background for the study is not clearly stated.

Author Response

RESPONSE TO THE REVIEWER 2

  • Description of the results is not clear. 65 eyes were studied, 43 served as control, thus it means that majority of studied eyes were control eyes. How control group was provided? Was scleral buckling surgery performed in normal eyes with retinal detachment? It should be clearly stated that control eyes were emmetropic eyes with retinal detachment. There were only 9 high myopic eyes and 13 low myopic eyes. These numbers are low. How did you assess the number of subjects in each group?

We studied a series of consecutive patients with rhegmatogenous retinal detachment (RRD) who underwent extraocular retinal detachment surgery in our clinic. Our aim was not, for this study, to specifically compare emmetropes/low and high myopic eyes.  In order to evaluate if there was changes in SRF composition related to refractive status, we grouped patients in emmetropes, low and high, as we separate for extension quadrants. We have change in the text in order to clarify and we have changed “control” by “emmetropic” to avoid confusing terms.

Material and Metods pag 2 line 8: that permed us to separate in (E) emmetropes with a spherical equivalent between 0 and +0.5 diopters, low myopia (LM) between 0 and -5.95 D and high myopia (HM) above -6 D”,

  • In the abstract conclusion is not supported by the results. It is not clearly described what group of patients are examined.

Pag 1 abstract line 10. We have modify the abstract to clarify, “Our results suggest that oxidative imbalance seems to be important in more extended RD and in myopic eyes and probably should be taken into account in the managing of this cases”. 

  • It is later written: “Detached retina extension was 19, 25, 13 91 and 8 eyes for one, two, three and four detached quadrants, respectively”. This sentence is not easy to be understood.

Results page 3, line 11 We have changed to: . “Detached retina extension was: 19 eyes had one quadrant detaches, 25 tow quadrants, 13 three quadrants and 8 eyes four detached quadrants”

  • Statistical differences were set at p ≤ 0.01 why not 0.05? We did the statistical study with a p at 0.05 and at 0.01. But since the significant differences, when there were any, reached p 0.01 in all cases, we established that level of significance that is more powerful. We have not explained on the text, we can do if considered necessary
  • Abbreviations should be described when describing figures (for example MDA in figure 1 description). Done
  • In the introduction the background for the study is not clearly stated. We have changed the redaction in this part of the abstract in order to make it more evident:

Pag 1 Introduction line 7 and following… “attributed this lack of total functional recovery after successfully retinal reattachment, to programmed apoptosis of retinal cells [4,5]. Oxidative damage is a well-known mechanism of apoptosis induction. Both have been frequently founded in ocular diseases [6]. , and iIt It is already known that the vitreous humor of patients with retinal detachment has oxidative products in it [7,8], although not specifically with lipid peroxidation products (LPO), which have a demonstrated high toxic activity [9,10];. Nevertheless, the nature of subretinal fluid (SRF) present in RRD has not been thoroughly studied. As SRF comes into direct contact with the full-of-cell membranes outer segments (OS) of the photoreceptors in the detached retina, lipid peroxidation research on this SRF can occupy a prominent place to study the pathogenesis of damage due to retinal detachment. Therefore, the relationship between lipid peroxidation and parameters identified as key in this disease such as age, refractive status, evolution time and extension of RRD has not been studied in RRD either [11].

Malondialdehyde (MDA) is one of the most well-known secondary products of lipid peroxidation (LPO), it has a half-life in physiological conditions very superiors to the extremely short half-lives of reactive oxygen species [12] and is a marker of peroxidative damage to cell membranes [13,14]. . The aim of this study was to identify a relation between clinical characteristics and differences in LPO in the SRF of retinal detached patients by MDA quantification.”

We have also developed a little more some ideas in the discussion in respect of what the reviewer suggests, to try to clarify them:

Discussion pag 5 lin 8 “MDA has a half-life in physiological conditions very superiors to the extremely short half-lives of reactive oxygen species (Esterbauer H, 1991). Once formed, MDA can be enzymatically metabolized (oxidation and decarboxylation to CO2 and H2O) or can react in vivo on cellular and tissular proteins and nucleic acids to form adducts. Far to neutralize its toxicity, these changes can induce either biomolecular damages or cell death (Ayala A, 2014; Gaschler MM, 2017), because apoptosis, autophagy and ferroptosis (Su L-J, 2019).

There are many toxic biological effects that they can cause on proteins (GÄ™gotek A, 2019) such as pro-inflammatory alterations of components of the complement system  factor H, main regulator of the alternative pathway (Weismann D, 2011) and C3a, proinflammatory complement component (Veneskoski M, 2011); activation of a specific isoform of protein kinase C (PKC) (Kharbanda KK, 2002); MDA adducts with eElongation factor 2 (eEF2), which catalyzes the movement of the ribosome along the mRNA in protein synthesis could contribute to decline of protein synthesis (Argüelles S, 2013);… Furthermore, MDA is an important contributor to DNA mutation and damage (Gentile F, 2017)”.

Also we have sent the manuscript to the English service

We thank the reviewer for his comments that have allowed us to improve the manuscript.

Round 2

Reviewer 1 Report

According to the reviewer’s comments, the authors responded clearly to them and rewrote the whole article. The reviewer made sure that these additional discussions improved the quality of this article.

Reviewer 2 Report

All comments have been adressed properly.